# Robust Beam Selection Based on Water Equivalent Thickness Analysis in Passive Scattering Carbon-Ion Radiotherapy for Pancreatic Cancer

**DOI:** 10.3390/cancers15092520

**Published:** 2023-04-28

**Authors:** Yuan Zhou, Makoto Sakai, Yang Li, Yoshiki Kubota, Masahiko Okamoto, Shintaro Shiba, Shohei Okazaki, Toshiaki Matsui, Tatsuya Ohno

**Affiliations:** 1Graduate School of Medicine, Gunma University, Maebashi 371-8511, Japan; 2Gunma University Heavy Ion Medical Center, Maebashi 371-8511, Japan; 3Department of Radiation Oncology, Harbin Medical University Cancer Hospital, Harbin 150040, China; 4Department of Radiation Oncology, Shonan Kamakura General Hospital, Kamakura 247-8533, Japan

**Keywords:** water equivalent thickness, pancreatic cancer, robust beam configuration, carbon-ion radiotherapy, accumulated dose distribution, angular dependency

## Abstract

**Simple Summary:**

Anatomical variations may distort the carbon-ion beam, leading to dose degradation during treatment. Thus, a robust-beam arrangement is important for patients with pancreatic cancer with carbon-ion radiotherapy. Our study aimed to investigate the angular dependency of water equivalent thickness (WET) variation for pancreatic cancer and to evaluate the robustness of accumulated doses with WET-based beam configurations. We found that posterior oblique beams in the supine position and anteroposterior beams in the prone position were the most robust to WET changes, and the robustness of the accumulated dose was significantly improved by using WET-based beam configurations. The findings may provide a reference for robust beam selection in clinical practice.

**Abstract:**

Carbon-ion radiotherapy (CIRT) is one of the most effective radiotherapeutic modalities. This study aimed to select robust-beam configurations (BC) by water equivalent thickness (WET) analysis in passive CIRT for pancreatic cancer. The study analyzed 110 computed tomography (CT) images and 600 dose distributions of eight patients with pancreatic cancer. The robustness in the beam range was evaluated using both planning and daily CT images, and two robust BCs for the rotating gantry and fixed port were selected. The planned, daily, and accumulated doses were calculated and compared after bone matching (BM) and tumor matching (TM). The dose-volume parameters for the target and organs at risk (OARs) were evaluated. Posterior oblique beams (120–240°) in the supine position and anteroposterior beams (0° and 180°) in the prone position were the most robust to WET changes. The mean CTV V95% reductions with TM were −3.8% and −5.2% with the BC for gantry and the BC for fixed ports, respectively. Despite ensuring robustness, the dose to the OARs increased slightly with WET-based BCs but remained below the dose constraint. The robustness of dose distribution can be improved by BCs that are robust to ΔWET. Robust BC with TM improves the accuracy of passive CIRT for pancreatic cancer.

## 1. Introduction

Carbon-ion radiotherapy (CIRT) is a potential treatment modality for unresectable pancreatic cancer because of its marked advantages in dose conformity [1] and relative biological effectiveness (RBE) [2]. Several studies have reported that CIRT achieves better clinical outcomes of overall survival and local control than conventional photon therapy [3,4,5]. However, despite the carbon-ion beam offering high dose conformity owing to its characteristics of Bragg peaks and sharper penumbra [1], rapid fall-off at the beam distal edge is also very sensitive to anatomical changes.

Previous studies have attempted to improve pancreatic tumor location reproducibility using tumor matching (TM) based on in-room computed tomography (CT) to mitigate dose degradation [6,7]. Even though TM improved the dose coverage in pancreatic cancer compared to ordinary bone matching (BM), the effect was not as pronounced as in lung and liver cancers [8,9], as the anatomical changes around the pancreas tend to change beam ranges [6,10,11,12]. Moreover, BM is still the mainstream matching method because many facilities do not have an in-room CT system, and CT scans reduce throughput and increase radiation exposure.

Determining the optimal beam arrangement that is robust to anatomical changes in pancreatic cancer could effectively mitigate dosimetric degradation [13]. In CIRT, the water equivalent thickness (WET) variation, as well as the beam axis, is an important parameter because it determines the range, thereby enabling the assessment of treatment effectiveness. WET variations can occur due to a variety of factors, including differences in tissue composition, density, tissue morphology, body weight, and/or movements of organs, such as the stomach and intestines. Understanding WET variations is important for optimized treatment planning and ensuring that the intended dose is delivered to the tumor while minimizing the dose to the surrounding normal tissue [14,15,16,17,18,19,20]. Several studies have investigated the angular dependency of the beam range variation using WET for thoracic cancer and head and neck cancer and have confirmed a strong correlation between dose reduction and WET variation [16,17,18,19,20].

Worst-case optimization is a possible option [21]. However, accurate simulation of the worst-case distribution is challenging because of unpredictable gastrointestinal (GI) changes. Although Dreher et al. [22] analyzed the dose variation in different beam configurations (BCs) and recommended a single posterior beam (180°) in the supine position, it is unrealistic, considering the dose to organs at risk (OARs), especially for the passive carbon-ion technique.

Pancreatic cancer treatment with CIRT is generally performed using two positions (supine and prone) to spare the OAR dose, even in facilities with gantry [3,4,5]. Considering that the robust beam range may vary with different treatment positions and matching methods (TM and BM), assessing the robustness in both positions with TM and BM is necessary.

Thus, this study aimed to investigate the angular dependency of WET variation in the supine and prone positions for pancreatic cancer and to evaluate the robustness of accumulated doses with WET-based BCs under two matching methods (BM and TM). To the best of our knowledge, this is the first study to evaluate robust BCs by calculating the WET variation in carbon-ion therapy for pancreatic cancer.

## 2. Materials and Methods

### 2.1. Patients and CT-Image Acquisition

This retrospective study evaluated eight patients with pancreatic cancer who underwent passive scattering CIRT. A total of 110 CT images that included 15 for treatment planning (CT_plan) and 95 on every treatment day (CT_daily) were studied. On each of the 12 treatment days, CT_daily images corresponding to the treatment position were taken after treatment irradiation under the same conditions as CT_plan (one CT_daily image was missing due to CT device failure).

In clinical practice, orthogonal radiographic images are taken around the peak exhalation phase for alignment with the BM prior to treatment. Generally, positioning with TM during treatment uses a CT image taken around the end of the expiration phase with a respiration monitor under free breathing before irradiation. However, in this study, BM was used as the actual treatment, and CT scans were performed on patients under free breathing after the treatment was completed. The gating levels for CT_plan and CT_daily were set at ±30% for treatment irradiation.

The voxel size of all the CT images was 1.07 × 1.07 × 2.00 mm. CT_plan images were taken at the end of expiration in both supine and prone positions. However, the plan_CT of patient 5 in the prone position was removed because the patient was treated only in the supine position because of the poor dose distribution. The plan_CT of the prone position of three patients was performed after the couch was rolled ±10° around the superior–inferior axis to spare the dose to the GI tract.

This study was approved by our facility’s institutional review board (approved number: 1564) and was conducted in accordance with the ethical standards of the institutional and/or national research committee and the 1964 Helsinki Declaration and its later amendments or comparable ethical standards. The study was registered in the University Hospital Medical Information Network Clinical Trials Registry (UMIN-CTR trial number: 000029495), and all study participants provided written informed consent.

### 2.2. Original Treatment Plan

Treatment planning with four-box fields (0°, 90°, 180°, and 270°), defined as the original beam configuration (BC_original), was performed using a XiO-N system (Elekta, Stockholm, Sweden, and Mitsubishi Electric, Tokyo, Japan). Passive scattering CIRT based on single-field uniform dose optimization for the target was performed using vertical and horizontal fixed beamlines [23], and only one field was delivered per fraction. Patients were treated in the supine position for days 1–9 and in the prone position for days 10–12. The target and normal tissue contours on the CT_plans and CT_dailys were delineated by an experienced radiation oncologist. The clinical target volume (CTV) for the initial nine fractions (CTV1) was delineated by an experienced radiation oncologist expanding the gross tumor volume (GTV) by at least 5 mm. As CTV2, the GTV was expanded by only 5 mm for three boost irradiations, excluding the GI tract. The planning target volumes (PTV1 and PTV2) were expanded by a 3 mm margin to the CTVs and were adjusted to avoid the GI tract. The detailed target volume is listed in Table 1.

The unit of carbon-ion dose was defined as the clinical dose, which was the RBE-weighted absorbed dose [24]. The administered dose of carbon ion was 55.2 Gy (RBE) for PTV2 in 12 fractions for 3 weeks (4.6 Gy (RBE) per fraction) and 41.4 Gy (RBE) for PTV1 in the first 9 fractions. Whenever possible, treatment planning was developed so that 95% of the prescribed dose covered at least 95% of the CTV (CTV V95 >  95%). The dose constraints for OARs were as follows: minimum doses to the highest irradiated 2 cubic centimeter volume (D2cc) of the stomach and duodenum <44 Gy (RBE); a mean dose (Dmean) of <15 Gy (RBE) to the ipsilateral kidney; and the maximum dose (Dmax) to the spinal cord limited to 30 Gy (RBE). The dose constraints for OARs were prioritized over the dose coverage for CTVs.

### 2.3. WET Analysis

An in-house program was developed to calculate the WETs from the body surface to the distal edge of the CTVs. WET analysis was performed for every 5° of the 360° coplanar direction. The impact of day-to-day anatomical changes between those in CT_plan and CT_daily was calculated on the WET for CTV1 and CTV2. Because the wobbler magnet was positioned 900 cm away from the isocenter, we ignored the deviation caused by the angles of the rays for simplicity. The workflow is as follows, and an example calculation at 0° is shown in Appendix A.

To identify beam paths passing through the CTV in all selected beam angles (0° to 360° in steps of 5°), raytracing was performed for each CT_plan and CT_daily [16,17,18,19,20]; the ray separation interval was equal to the voxel size (i.e., 2 mm in the SI direction and 1.07 mm in the coronal plane) [see Appendix A];CT_daily was aligned rigidly to the corresponding CT_plan using BM or TM, wherein the CT_plan of the supine and prone positions were aligned to the corresponding CT_daily;The total WET value from the body surface to the distal end of a CTV in a given beam path was the summation of the WET of all voxels along the path. The WET of a voxel was calculated as the product of the intersection length of the path within the respective voxel and the stopping power ratio evaluated from the CT value. The WET values were calculated in the CT_plan (WET_plan) and CT_daily (WET_daily) after BM and TM, respectively [Appendix A];The paths that simultaneously passed the CTV_plan and CTV_daily were identified. The WET change of each interested path was calculated as the WET_plan minus the corresponding WET_daily [Appendix A];ΔWET was averaged for the absolute WET change of all identified paths [Appendix A].

The formula for ΔWET calculation is as follows:$$\Delta WET=\frac{1}{n}\;\sum_{n}\left|WET\_plan-WET\_daily\right|$$
where *n* is the number of identified paths in one CT image at one beam.

For each angle and for each patient, the number of identified paths was beyond 1000.

The angular sensitivity of ΔWET differed due to differences in the treatment positions (supine and prone) and matching methods (TM and BM). Thus, we calculated the ΔWET in both treatment positions (ΔWET_SP and ΔWET_PP) after TM or BM.

### 2.4. ΔWET Based Beam Configuration and Planning

In this study, the definition of the directions of rotations was defined according to IEC, in which the frontal (anterior) direction of the body was set at 0°, and the left-hand side of the body was set at a positive angle. We defined two ΔWET-based BCs: for the rotating gantry (BC_gantry) and the fixed port (BC_fixed) (Figure 1e,f). To protect OARs, the ΔWET-based BCs included supine and prone positions. For BC_fixed, the beam selection range was set to within ±20° of the top-to-bottom or bilaterally horizontal direction, considering the couch rotation limit and the dose constraint for OARs. The dose to normal tissue was minimized by irradiation from four directions. However, despite being helpful in suppressing the dose to the stomach, the prone position was only adopted for one field due to the increased burden on the patient. Based on the selectable angles with minimal ΔWET, the angles selected for BC_fixed were 355°, 110°, and 255° in the supine position and 180° in the prone position. For BC_gantry, the initial angle chosen was 210° in the supine position, corresponding to the minimum ΔWET with the TM. To avoid the spinal cord and right kidney while minimizing the ΔWET, 150° and 0° were selected in the supine and prone positions, respectively. Moreover, to further alleviate the significant burden of the prone position treatment on the patients and to minimize the stomach dose, 345° was added to the supine position. A total of 69 planned doses with three BCs were evaluated.

Because only limited data were available for this study, ΔWET was calculated using all data, and robust angles were selected. The same data were used in the subsequent dose distribution analysis to confirm the robustness of the dose distribution. This has the risk of introducing the problem of self-referentiality. To address this issue, we conducted a preliminary evaluation of the robust beam range using a cluster of six randomly selected patients from the eight included in the study. We evaluated the robust beam range by calculating the average change in ΔWET in this cluster. This process was repeated 17 times, and we obtained consistent results each time (Appendix A).

### 2.5. Daily Dose Evaluation

The dose distributions for each treatment day were recalculated based on the corresponding CT_daily after being aligned with the corresponding CT_plan by TM or BM. A total of 552 daily doses of BM or TM were evaluated. To assess the robustness of the treatment plan, we calculated dose reduction, which is the difference between the planned and daily doses. Because the patient remained in just one position during daily CT scanning, the daily dose could only be evaluated for that specific position. The treatment plan involved irradiation from three directions while the patient was in the supine position; the effect of each day’s patient-specific changes on the dose distribution depended on the irradiation angle. Limited data were collected to reduce the effect of the irradiation schedule; thus, to achieve a more significant evaluation with limited data, we calculated the total doses of the beams from all three directions for all CT_daily scans. Consequently, the dose for each day was tripled for the evaluation in the supine position; thus, the dose was set to one-third. Each dose reduction was evaluated as a percentage of the prescription dose per fraction (4.6 Gy).

### 2.6. Accumulated Dose Evaluation

To calculate the accumulated dose distribution, the daily dose distributions (4.6 Gy (RBE) per fraction) on each CT_daily were warped and transferred to the corresponding CT_plan by hybrid-DIR using MIM Maestro (MIM Software, Beachwood, OH, USA). The accuracy of the hybrid-DIR was confirmed in a previous study [25]. We set the beam schedule for two ΔWET-based BCs referring to the original plan (Appendix A). Then, the deformed doses of the total 12 fractions were accumulated in the CT_plan according to the beam schedule (48 accumulated doses). To confirm the effect of the beam schedule, dose differences with various beam schedules were calculated (Appendix A).

### 2.7. Statistical Analysis

The normality of the distribution of the selected comparisons was evaluated using the Shapiro–Wilk normality test. The significance of statistical differences was analyzed using the paired *t*-test for normally distributed data and the paired Wilcoxon signed-rank test for non-normally distributed data. In addition, the Fligner–Killeen nonparametric test was used to test the homogeneity of variances of daily dose differences of OARs. Statistical significance was set at 0.05, and the *p*-values were adjusted for multiple testing using the false discovery rate method. All statistical tests were performed using R-4.2.2.

## 3. Results

### 3.1. ΔWET Analysis

A similar angular dependence of WET change was observed in most CT_daily images in the same treatment position, regardless of the use of TM and BM (Figure 1). Posterior oblique (120–240°) beams in the supine position and anteroposterior (0° and 180°) beams in the prone position were the most robust to WET variation. The mean value ± standard deviation of ΔWET overall beam angles with BM and TM in the supine position ranged from 4.8 ± 2.2 to 11.4 ± 5.3 mm and from 4.3 ± 2.2 mm to 11.4 ± 5.3 mm, respectively. In the prone position, it ranged from 5.2 ± 1.5 mm to 12.7 ± 5.3 mm and from 4.7 ± 1.6 mm to 12.9 ± 5.9 mm, respectively.

### 3.2. Daily Dose Variation

Figure 2 and Figure 3 show the difference in DVH (dose–volume histogram) for the target and OARs in the daily dose, respectively. The dose reduction for CTVs with both proposed BCs was significantly lower than for those with BC_original, especially for TM. The dose for one patient (patient 5) was significantly lower than those for the other patients, especially with BM in the supine position. This could be attributed to significant tumor migration. For this patient, ΔWET was very large in any direction, and it was difficult to obtain a good dose distribution for both BM and TM.

For OARs, the difference in the dose variation with the three BCs is very slight in the supine position. In the prone position, the dose variation of D2cc of the stomach with BC_gantry with TM was significantly better than BC_original/BC_fixeds.

The analysis of the homogeneity of variation showed that in the supine position, only those in both kidneys were significant. For the prone position, the variance of the dose variation with BC_gantrys in D2cc of the stomach is significantly smaller (more robust) than with BC_original.

### 3.3. Accumulated Dose Variation

An example of the planned and accumulated doses is shown in Figure 4. The planned WET-based BCs could maintain good dose coverage with reference to BC_original. Figure 5 shows the DVH parameters for the planned and accumulated doses. The accumulated doses with both proposed BCs with BM had significantly higher D95% (the dose delivered to 95% of the volume) of CTV than those with BC_original, and for TM, CTV D95% with BC_gantry was significantly higher than with BC_original. Moreover, the mean reduction in CTV V95% with BM was significantly lower in both proposed BCs than with BC_original (adjusted *p* < 0.05), which were 10.3%, 11.4%, and 16.6% for BC_gantry, BC_fixed, and BC_original, respectively.

For OARs, the D2ccs of the stomach and duodenum and the Dmax of the spinal cord with both proposed BCs were higher than those of the BC_original in some patients. However, the difference was due to the planned dose distribution, and there was no statistical difference in the magnitude of variation or its deviation, which indicates robustness.

Additionally, analysis with various beam schedules resulted in only small differences in the integrated dose (Appendix A).

## 4. Discussion

In this study, we considered a robust treatment plan using CIRT in pancreatic cancer. We propose robust BCs (for fixed ports and gantry systems) based on an analysis of WET. Because changes in WET may not accurately predict changes in dose distribution, the effectiveness of robust beam optimization was evaluated by comparing dose distribution changes with robust BCs to those with conventional BC.

We considered the robust angles based on the change in WET because we believe that WET variation has a particularly large impact on dose distribution. During the evaluation, changes in WET were calculated only for beam paths passing through both CTV_daily and CTV_plan. With this approach, the tumor motion could reduce the overlap between CTV_plan and CTV_daily, potentially leading to errors in assessing robustness. However, the proportion of overlap was not included in the evaluation in this study because the ratio of the overlap (i.e., the ratio of the beam paths passing through CTV_daily to those that also pass through CTV_plan) did not improve the ability to estimate the robustness of dose distribution in the preliminary study. Furthermore, there was almost no difference in the robustness direction between TM and BM, although TM can suppress the decrease in the ratio of overlap due to tumor migration compared with BM because TM moves the beam axis according to the migration. This result suggests that the decrease in the overlap ratio due to tumor migration has a small impact on the consideration of robust irradiation angles. Because ΔWET would also be larger if the tumor moved, it may have been sufficient to evaluate robustness.

Our study findings showed that WET changes were small in several directions. For example, posterior-oblique beams in the supine position had high robustness with respect to the WET variations in the CT_dailys, which was in agreement with a study by Batista [26]. However, for the posterior oblique angles in the prone position, the WET changes were not as robust as those in the supine position. This is due to the compression of the bowels, causing them to shift toward the sides and posterior oblique direction and enter the beam range of the posterior oblique beam. This resulted in a noticeable change in the WET of the posterior oblique beam in the prone position due to the change in gas in the bowels. On the other hand, a few results contradict the suggestions of previous studies. Some studies [22,26] have shown that anterior beams may not be robust due to the GI tract; however, our findings showed that the anterior beams have a relatively small ΔWET and were thus adopted for the BCs. A comprehensive evaluation of the angular dependence of ΔWET revealed that anterior beams may undergo a lesser WET change than lateral and frontal oblique beams. This may be due to the relatively small volume of the GI tract that the beam passes through. Furthermore, beams from the frontal side of the body were more robust in the prone position than in the supine position; this is because most of the gas in the stomach is squeezed out of the field in the prone position, thereby mitigating the dose variation owing to the volume change of gas in the beam path. Similar phenomena have been reported earlier by Miki et al. [27].

Nevertheless, although we selected the beam angle with the minimum mean ΔWET, the ΔWET with BM exceeded the maximum PTV margin (5 mm) in the prone position. This suggests that using the BM for a 5 mm PTV margin may be risky, particularly in the prone position.

For proposed robust BCs, some DVH indices for the OARs were higher than those for the original treatment plan because the shape of a spread-out Bragg peak caused a poor dose distribution to normal tissue at the proximal side of the target. However, the values of the indices were within the dose constraints for all treatment plans. Moreover, our proposed BCs met the constraints for OAR in the accumulated doses in all patients except for patient 5, which was also true for the BC_original. For TM, except for patient 5, there was only a single patient whose Dmax of the spinal cord of an accumulated dose with BC_gantry was slightly (31.3 Gy) larger than 30 Gy.

We also found a highly robust dose distribution of BC_gantry with TM, and the dose degradation in CTV V95% was very limited (mean value: 3.8%), which was close to that of the adaptive method (mean value: 2.1%) reported in a previous study [28]. Adaptive therapy has been verified as the most effective method for improving the robustness of CIRT plans [28,29,30]. However, rapid online dose verification and quality assurance systems are still challenging. Additionally, some treatment centers lack an in-room CT for the evaluation of anatomical changes, making it more challenging to perform adaptive CIRT. Considering the current situation of technique development, our method seems more practical and easier to use clinically than adaptive therapy.

The dose reduction, even with BC_gantry, was not sufficiently compensated for the accumulated dose in patient 1, possibly because the PTVs did not have enough margin for the CTV on the distal edge of the posterior angle to spare the stomach, and an anterior beam (selected in the supine position for BC_gantry) led to a poor distribution on some daily CTs.

Most patients exhibited similar patterns in the angular dependence of ΔWET. However, significant tumor displacement in patient 5, the CTV_daily deformed noticeably, resulting in an increase in WET variation at all angles and a weakened angular dependence of ΔWET, especially when using BM. This led to significant decreases in the daily and accumulated doses for all three BCs (Figure 2 and Figure 5), particularly when using BM; the daily doses for all three decreased by over 40%. If tumor migration is too large, the dose distribution may be significantly reduced, even if TM is used. In such cases, re-planning may be a better alternative.

Pancreatic cancer is challenging to treat due to its complex location; it is surrounded by OARs and is susceptible to structural changes in the body. Therefore, many studies have been conducted on robust treatment planning. Worst-case optimization and robust planning are methods for improving treatment planning while focusing on changes in WET [31,32,33]. These methods can optimize the treatment plan to suppress changes in dose distribution, even when changes occur within the beam range. However, because they assume uniform changes in WET and tumor position, it is difficult to incorporate large partial changes in WET due to irregular bowel movements. It is also difficult to conduct an exhaustive study for robust angles. In our study, we evaluated the changes in WET at each angle. By using robust angles and incorporating the possible WET changes for each angle into worst-case optimization, more robust planning can be achieved.

This study has some limitations. First, the use of biological dose accumulation may introduce inconsistencies and limitations in the evaluation of dose effectiveness [34]. Second, although the plan with BC_fixed with TM showed worse robustness than the plan with BC_gantry, the results were acceptable for the passive scattering CIRT for most patients (mean dose reduction: −5.22%) compared with the conventional method. However, it should be noted that BC_fixed needs to be achieved by rotating the treatment couch, which may cause changes in anatomic structure and thus affect the robustness of the beams. ΔWET-based BC would vary depending on the planning techniques (i.e., passive or scanning, fixed-port, or gantry-system) and characteristics of dose-blurring. One study [35] demonstrated that good dose distribution can be achieved with three supine posterior fields (150°, 180°, and 210°) using a scanning technique. Thus, a robust BC in which beams are selected with minimal WET variation may be feasible for the scanning technique [32]. The robust beam range in this study would be indicative for most current particle radiotherapy techniques. Moreover, our study did not investigate intrafractional anatomical changes, such as respiratory and gastrointestinal (GI) motion [10,36]. A respiratory-gating system would reduce respiratory motion, but it is not completely negligible. The target coverage should be worse than what was observed in our study due to respiratory motion and other intrafractional changes. However, our method could help to improve the robustness against intrafractional changes caused by a gas motion in GI because the motion pattern should be similar for both intra- and interfractional changes.

Finally, considering the small sample size of this study, it is important to assess the potential for bias when evaluating the robust beam range by ΔWET analysis. Although several evaluations were conducted in our study to increase the statistical significance (as shown in Appendix A), larger studies are necessary to confirm these results and increase the generalizability of the findings.

In summary, our research, combined with larger subsequent studies, has the potential to significantly impact clinical treatment planning workflows, especially for beam arrangement selection, for pancreatic cancer with CIRT.

## 5. Conclusions

We investigated anatomically robust BCs based on WET variations for pancreatic cancer treatment with CIRT. A significant angle dependency on interfractional change was observed. For most of the patients, posterior oblique beams (120–240°) in the supine position and anteroposterior (0° and 180°) beams in the prone position were more accurate for minimizing the interfractional change than the other beams. Additionally, treatment plans with WET-based BCs were the most robust in the majority of patients. However, the improvement in robustness with BM was limited owing to significant tumor movement in some patients.

## Figures and Tables

**Figure 1 cancers-15-02520-f001:**
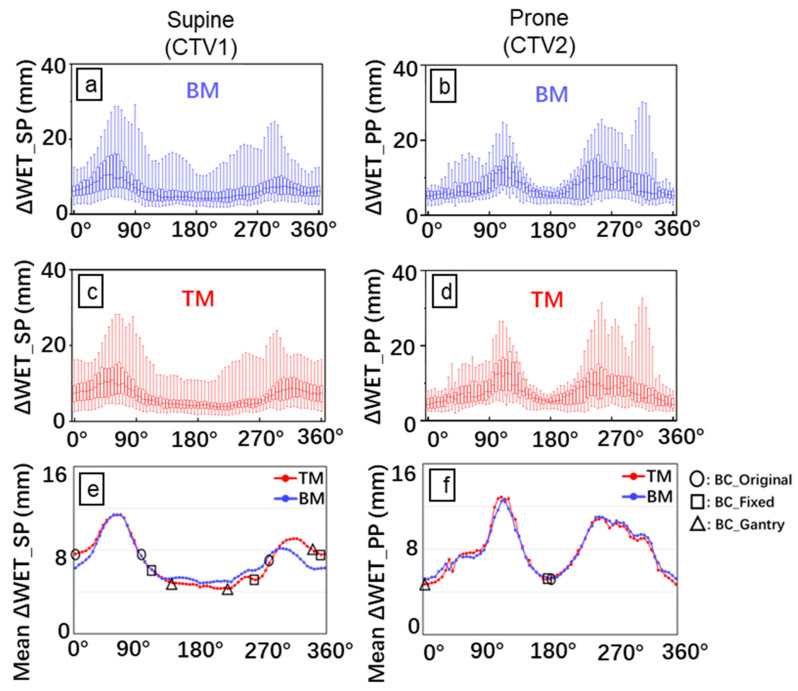
Analysis of water equivalent thickness (WET). Box plots of the absolute WET difference with bone matching (BM) (**a**,**b**) and tumor matching (TM) (**c**,**d**) for all CT_daily images. The mean ΔWET (**e**,**f**) for the eight patients was determined by plotting ΔWET against the beam angle. The ΔWET_SP and ΔWET_PP correspond to the WET difference in the supine (left side) and prone (right side) positions, respectively. The X-axis is the beam angle with an interval of 5°. The empty square indicates the BC_original; empty circles, BC_fixed; and empty triangle, BC_gantry. Blue line and box: BM; Red line and box: TM.

**Figure 2 cancers-15-02520-f002:**
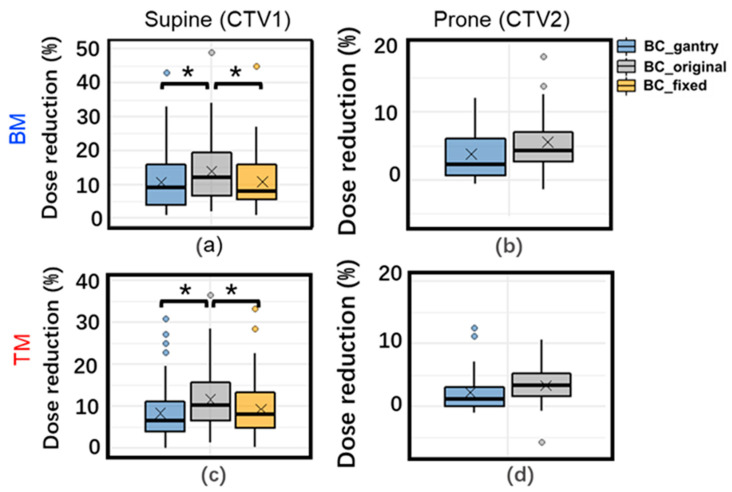
Box plot of daily dose variation for clinical target volume (CTV) V95% with three beam configurations with BM (top) (**a**,**b**) and TM (bottom) (**c**,**d**) in supine (left) and prone (right) positions. *: Adjusted *p* < 0.05 for tests of difference in means. ×: mean daily dose variation. The data with BC_fixed in the prone position has been omitted, as BC_fixed in the prone position is the same as BC_original.

**Figure 3 cancers-15-02520-f003:**
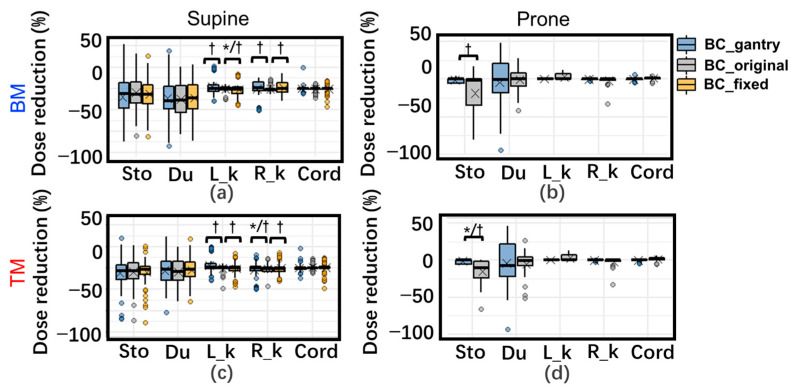
Box plot of daily dose variation for organs at risk (OARs) with three beam configurations (BCs) with BM (top) (**a**,**b**) and TM (bottom) (**c**,**d**) in supine (left) and prone (right) positions. Dose variation (%) means the dose variation as a percentage of the corresponding prescription dose. *: Adjusted *p* < 0.05 for tests of difference in means. †: Adjusted *p* < 0.05 for tests of deviation. ×: mean daily dose variation. Abbreviations: Sto: D2cc of the stomach; Du: D2cc of duodenum; R_k: Dmean of Right_kidney; L_k: Left_kidney; Cord: Dmax of spinal cord. The data with BC_fixed in the prone position has been omitted because BC_fixed in the prone position is the same as BC_original.

**Figure 4 cancers-15-02520-f004:**
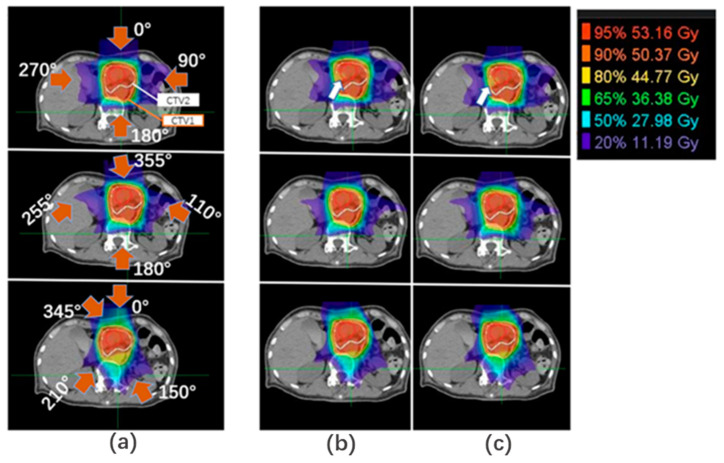
An example of dose distribution. Dose distributions of (**a**) planned dose and accumulated dose with (**b**) BM and (**c**) TM. Red arrows: beam direction; white arrows: coverage reduction; red lines: gross tumor volume (GTV); white lines: CTV2; orange lines: CTV1.

**Figure 5 cancers-15-02520-f005:**
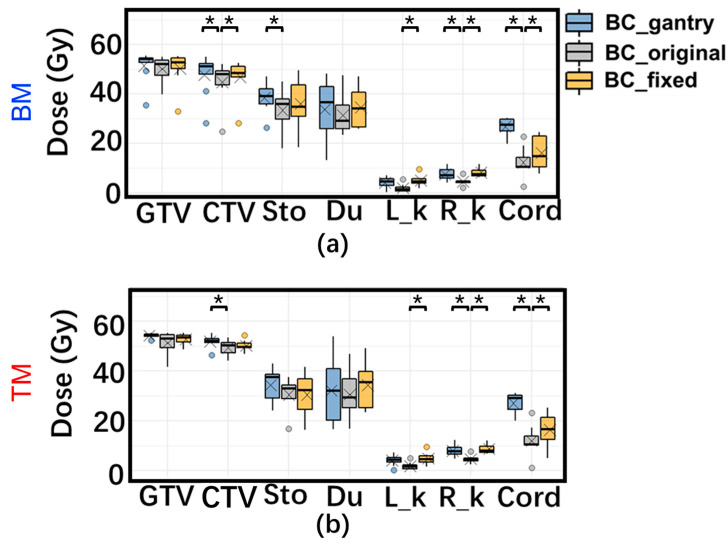
Box plots of planning dose and the accumulated dose difference of three BCs with (**a**) BM and (**b**) TM for all patients of DVH. ×: mean daily dose variation. Abbreviations: GTV: D95% of GTV; CTV: D95% of CTV; Sto: D2cc of the stomach; Du: D2cc of duodenum; R_k: Dmean of Right_kidney; L_k: Left_kidney; Cord: Dmax of the spinal cord. *: Adjusted *p* < 0.05 for tests of difference in means.

**Table 1 cancers-15-02520-t001:** CT number and target characteristics.

Patient Number	Age	Sex	Target	CTV_Plan Volume (cm^3^)	CTV_Daily Volume (Mean ± SD) (cm^3^)	
1	50	F	CTV1:	94.0	99.3	±4.8
			CTV2:	51.1	53.0	±1.9
2	76	M	CTV1:	158.1	156.7	±5.3
			CTV2:	63.6	57.0	±3.9
3	83	F	CTV1:	170.3	173.9	±6.2
			CTV2:	89.4	76.1	±3.1
4	81	M	CTV1:	202.0	176.4	±15.2
			CTV2:	113.4	83.4	±4.7
5	51	F	CTV1:	126.8	114.6	±9.5
			CTV2:	54.5	48.7	±4.8
6	61	F	CTV1:	127.6	126.5	±7.0
			CTV2:	70.3	55.8	±7.6
7	78	F	CTV1:	131.3	129.0	±5.5
			CTV2:	45.8	52.9	±1.9
8	74	M	CTV1:	165.8	168.3	±7.0
			CTV2:	99.3	87.6	±1.8

One CT image of patient 7 is missing due to CT device failure. Abbreviations: CT, computed tomography; CTV, clinical target volume; F, female; M, male; SD, standard deviation.

## Data Availability

The datasets in this study are available from the corresponding author upon reasonable request.

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
