# Peer review of "Robust Beam Selection Based on Water Equivalent Thickness Analysis in Passive Scattering Carbon-Ion Radiotherapy for Pancreatic Cancer"

_cancers, 2023, doi:10.3390/cancers15092520_

Round 1

Reviewer 1 Report

The manuscript by Zhou et al. investigates the strategy to select treatment directions in carbon ion radiotherapy of pancreatic cancer based on the daily variations of the beam range to the target. The study is written mostly very well, and presents results of relevance for carbon ion therapy. However, the use of posterior oblique fields for pancreas is already quite established, and the differences in angle selection seem incremental to previous works, including study by some of the authors investigating adaptive beam angle selection strategies. I am therefore not convinced the study is of high enough relevance for the audience of Cancers. I list my detailed comments below.

Major comments:

1. Posterior oblique beam directions have been already identified in past literature to lead to stable beam ranges (see Batista et al. ref 25). The novelty this manuscript brings is to select the beam directions based on WET changes between daily CTs, and that it assesses two different rigid registration techniques. However, I personally think the beam angle selection should be done based on dose calculations and DVH metrics, rather than just the WET change, since that does not include OAR doses. For the pancreas, WET changes may be minimal but still considerable motion may be present, as reviewed in the "General IMPT treatment planning considerations" section in Tryggestad et al. (2020; 10.21037/jgo.2019.11.07).

The authors seem to also have added dose calculations in the BC_gantry case, where the beam angles were selected "to meet the above dose constrains for OARs" (Line 163). Is there a reason, aside from computation time, that favors using the WET changes over full dose calculations? Calculation speed in a clinical workflow, which is what the authors used to differentiate themselves against a previous article from their group that was looking into adaptive angle selection strategies, is not an argument, since this is a retrospective study, and using the dET in an adaptive strategy would still require dose calculations (see also comment 2).

2. Following the previous comment, it is not quite clear to me, how the authors decided on the gantry beam configurations. I assume a set of candidate directions was selected from the dWET plot and plan optimizations determined the OAR doses? What criteria were applied to determine the angular distance between the gantry angles selected? Could the authors provide more reasoning from the anatomical perspective for their choice of beam angles?

3. A 0 degree beam angle in the prone position enters right through the spine, which does not seem ideal when there are other options available. So I am wondering why the authors selected that direction over the 180 degree one they chose for the fixed beamline in the prone position?

4. The article does not discuss the issue of target motion during treatment. Motion during the irradiation can have quite a significant impact, especially for beams stopping right at critical structures, like the duodenum. From past studies put forward by some of the authors I take that they assumed a gated delivery, for which in their previous work was used to argue that motion impact was minimal. This should be mentioned in this article as well. Also, I believe the impact of motion on the choice of treatment angles needs to be at least discussed in this work. Using dWET changes for beam angle selection under motion has recently been highlighted in an AAPM task report to be useful for targets subject to intra-fractional motion (https://doi.org/10.1002/mp.15470), but it may not be ideal for the pancreas (see reference in comment 1).

I also would expect anterior beams to be quite affected by bowel gas movements in the intestines, which tend to be irregular and hard to predict. This has been pointed out as an issue in previous literature (e.g. Batista et al. Ref 25 or Kumagai et al. Ref 10), and could be a reason against anterior beam directions like the ), 355 and 345 degree directions selected in the present manuscript. Could the authors discuss this?

5. Line 145: It seems, the authors discarded those beam paths that did not crossed the CTV_daily anymore from the analysis. Wouldn't those beams also indicate an inferior robustness of that treatment angle? Was the ratio of beam paths that passed both the CTV_plan and CTV_daily constant over all beam angles? If not, what was the variation?

Minor comments:

1. In Figure 1, the markers for BC_Original and BC_fixed are swapped, when compared to the stated beam directions in the text and S2.

2. In Figure S1, the BM and TM WET images are the same, while the dWET images are not.

3. In Table S2, in the beam angles given for BC_original under Beam Configurations, 0 90 and 180 should be 0 90 and 270.

4. Line 173-179: This sentence is very hard to read, and should be split into at least two shorter sentences.

5. Line 158: For generality, the authors could refer to the IEC gantry coordinate system (see IEC 61217).

6. Line 90: the sentence starting with "but ..." is a repetition of a previous statement, and the but should be upper case

7. Was the dose accumulation one in biological doses? There are known inconsistencies in biological dose accumulation, pointed out for example in Niebuhr et al. (2021) https://doi.org/10.1186/s13014-021-01789-3 that the authors may find relevant for their work.

Reviewer 2 Report

Brief summary:

The paper analyzes the variability in water equivalent depth of the distal edge of the tumor depending on the beam angle selection in passively scanned carbon ion therapy of pancreatic cancer. Based on eight patients, it demonstrates that some beam angles are less affected by interfractional changes than others, which results in better dose distributions compared to a standard beam configuration of simple horizontal and vertical beams. This is an important finding for the treatment planning in patients, especially because it is the first study of this kind for pancreatic cancer.

General concept comments:

The study has one major design issue, that also was not discussed. The analysis of the dose distributions and the selection of beam angles is self-referential. The authors first analyze the deltaWET over all fractions of all patients. Then they select beam angles with minimal delta WET. And then they calculate the dose distributions with those angles on the same imaging data. If this work would be about an AI method, one would say that the training data is also the validation data. It is true that the authors can claim that if you would have selected those beam angles at the beginning of the treatment, it would have turned out to be a good choice as they didn’t observed severe range changes for those treatments. But if there would have been one in their data, they would have seen it in their deltaWET analysis and therefore wouldn’t have selected this angle. In the clinical situation, however, you do not have the daily CTs at the stage of treatment planning, so you need to estimate a good choice based on the experience from other patients.
The authors do not claim that their specific selection of beam configuration is the best for all patients, but they present a method how to find a good beam configuration. This could be done better by slightly modifying the study design to account for the mentioned bias.

    1. Define a procedure how to select beam angles based on a deltaWET analysis. (The authors did this already)

    2. For each patient, make a deltaWET analysis and select the beam angles according to step 1.

    3. Calculate the accumulated dose with the selected beam angles.

    4. Show that with your angle selection method, you get reasonable results.

    5. Now, that you proofed that you have a reasonable method to find good beam angles for new patients based on old patients, you can do what you show in your study: You select beam configurations based on your entire data set. You can reevaluate the doses with this selection as a final cross check. Ideally, the beam angle selection or treatment quality also does not change much compared to step 2.

Such a restructuring of the study might not be necessary if the authors provide good reasons why the explained bias is expected to be not severe. This could for example be, if the angle dependence of the deltaWET depends only little on the patient in their study.

Specific comments:

  1. Line 24: „Carbon-ion radiotherapy (CIRT) is the most advanced radiotherapy modality.” This is a very strong statement and the metric is very unclear. Photons have more patients, scanned ion beams can deliver more conformal dose distributions than the passively scattered ion beams you considered in your study. Please consider another sentence.

  2. Line 90-91: “but plan_CT of patient 5 in the prone position was removed because the patient was treated only in the supine position because of the poor dose distribution” This information was already mentioned in line 83-85). Please remove it ones.

  3. Line 123: “2 cubic centimeter area”. You mean volume, I guess.

  4. Line 161-163: Please explain in more detail how you selected the angles for the gantry. I understand that for BC_fixed, you selected the minimum of deltaWET within +-20° around the original positions. So deltaWET apparently is smaller for 355° than for 345°. So why did you then chose 345° for the gantry?

  5. Line 175: I do not understand this normalization. If I understand you correctly, you calculate D_total, normalized = (field 1 + field 2 + field 3) / 41.4 Gy +field 4 / 13.8 Gy. This means that you give a three times higher weight to the field in the prone position even though all fields were optimized to the same dose of 4.6Gy and delivered the same amount of time. Also with your formula, the total dose is 3 repetitions/field*[(3 fields supine *4.6Gy)/41.4Gy + (1 field prone * 4.6Gy)]/13.8 Gy=200%

  6. Line 188: “Randomly modified schedules”: The order of three fractions of one field before switching to the next field does not look random to me.

  7. Line 189: “Analysis”, not “Analyses” (typo)

  8. Figure 1: Did you accidentally swap the legend entries for BC_fixed and BC_original? Is the average profile representative for all patients or are the profiles totally different from each other?

  9. Line 218: How is the dose reduction defined? Is it the CTV V95% reduction compared to the planned dose distribution?

  10. Line 224: Is “daily dose variation” the same as “dose reduction”? If yes, please use same word.

  11. Discussion: Please address the problem described in the general concept section.

Reviewer 3 Report

The authors analyse robustness in water-equivalent thickness for a number of beam arrangements in carbon ion therapy of pancreatic cancer. The manuscript is well written, and methods & results seem sound. I have a few questions that need to be a addressed before publication can be recommended.

Introduction:

- Lines 57 to 64 are a little confusing. First, the authors talk about a beam arrangement, then directly about worst-case optimization (while the study [14], which they cite, does not really contain worst-case optimization on beam angle selection, but rather robustness analysis of beam arrangements. I don't understand the sentence after [14] (starting line 59). The proposed method is difficult or the simulation is difficult? 

-  I also think that there's more literature to worst-case/probabilistic optimization for CIRT, very recent paper 10.1016/j.radonc.2022.09.005, 0.3389/fonc.2022.974728, etc. for example. Please extend your literature review a little to cite relevant publications here.

Methods:

- The description of computing the WET would benefit by the use of the appropriate terms for the algorithms and could then be shorted. What the authors do is "raytracing" through a 3D voxelized geometry, algorithms for this are described in literature (and should probably be read/cited to make sure the computation is correct - as of now, the in-house implementation here is a blackbox for the reader and can not be validated). Thus, "Raytracing through the CTV was performed for all selected beam angles (...), with a ray separation of ..." sounds more appropriate in my opinion. 

- I have a small doubt about the selection of rays for raytracing. Are all rays chosen to be parallel? depending on the (virtual) source distance, non-parallel rays would be more appropriate in to model the geometric widening. Will probably not change the result much, but it would also be no problem for a raytracing algorithm to trace non-parallel rays (if that was indeed how they were chosen).

- Line 142: "length of the given path" is ambiguous (as there's no path "given" here). I guess it means the intersection length of the path/ray within the respective voxel, I hope? 

Minor Comments:

- Line 59 misses a period after citation [14].

Round 2

Reviewer 2 Report

General:

The quality of the paper was clearly improved and most of my questions where appropriately addressed by the revision.  But there a few comments left. The authors could  claim more explicitely the impact of their research (+ a follow up study with more statistics) on clinical treatment planning workflows.

Specific comments:

Line 167: The formula is missing in this PDF. Maybe just a rendering issue?

Line 190: I still don’t understand your selection criteria for the gantry configuration. There is also a grammar error here. I understand why you select 210° for supine and 0° for prone, as they are the minima of the deltaWET. But why did you chose exactly 150°? Did you have a criterion of at least 60° between fields? And if yes, why is it okay to have the two fields at 345° and 0° with only 15° gap in between?

Line 306:

Thank you for providing the new figure, which is really helpful. However, your statement, that you ‘applied the same method to select robust angles in this subset’ is not completely correct. You analysed the deltaWET again, but you did not reselect the beam angles. If I was you, I would move Figure S2 or at least the first reference to it to the methods section. I also think that you can discuss it in some more detail. For example, I see the pretty interesting result, that the deltaWET for some angles varies pretty much in the prone position, but it looks more stable in the supine position.

line 372: The sentence 'Pancreatic cancer is challenging to treat due to the complex location of the affected site' comes twice

line 402: 'If robust beams can optimize the uncertainty caused by GI motion during interfraction, they may also work similarly for intrafractional changes.' I strongly disagree to this statement. For interfractional motion, the water-equivalent distance between Bragg peaks remains constant. For intrafractional motion, changes between the delivery of subsequent slices change the overlap of the Bragg peaks of subsequent energy slices. In addition, the slices are moved lateral to the beam, causing more interplay. It is not as bad for passively scattered beams as for scanned ions beams, but still a severe issue that has to be properly discussed. Even for a gated delivery, the respiratory motion is not completely negligible, at least for the dose distribution in single fractions (it might after adding the doses of several fractions).

line 420: Figure S2 is not mentioned here.

Discussion/Conclusions: Please add a few lines in which you explicitly state the potential impact of your work on clinical treatment planning processes.

Reviewer 4 Report

I thank the authors for commenting the proposed issues. I would reccomend the pubblication of the paper in the present form.

Author Response

Thank you for your support and feedback.